# Synergistic Responses of Tibetan Sheep Rumen Microbiota, Metabolites, and the Host to the Plateau Environment

**DOI:** 10.3390/ijms241914856

**Published:** 2023-10-03

**Authors:** Yuzhu Sha, Xinyu Guo, Yanyu He, Wenhao Li, Xiu Liu, Shengguo Zhao, Jiang Hu, Jiqing Wang, Shaobin Li, Zhidong Zhao, Zhiyun Hao

**Affiliations:** 1College of Animal Science and Technology/Gansu Key Laboratory of Herbivorous Animal Biotechnology, Gansu Agricultural University, Lanzhou 730070, China; shayz@st.gsau.edu.cn (Y.S.); guoxinyu6662021@163.com (X.G.); zhaosg@gsau.edu.cn (S.Z.); huj@gsau.edu.cn (J.H.); wangjq@gsau.edu.cn (J.W.); lisb@gsau.edu.cn (S.L.); zhaozd@gsau.edu.cn (Z.Z.); haozy@st.gsau.edu.cn (Z.H.); 2School of Fundamental Sciences, Massey University, Palmerston North 4410, New Zealand; y.he@massey.ac.nz; 3Academy of Animal Science and Veterinary Medicine, Qinghai University, Xining 810016, China; 13893393775@163.com

**Keywords:** Tibetan sheep, rumen, microbial, metabolites, host, serum

## Abstract

Plateau adaptation in animals involves genetic mechanisms as well as coevolutionary mechanisms of the microbiota and metabolome of the animal. Therefore, the characteristics of the rumen microbiome and metabolome, transcriptome, and serum metabolome of Tibetan sheep at different altitudes (4500 m, 3500 m, and 2500 m) were analyzed. The results showed that the rumen differential metabolites at 3500 m and 4500 m were mainly enriched in amino acid metabolism, lipid metabolism, and carbohydrate metabolism, and there was a significant correlation with microbiota. The differentially expressed genes and metabolites at middle and high altitudes were coenriched in asthma, arachidonic acid metabolism, and butanoate and propanoate metabolism. In addition, the serum differential metabolites at 3500 m and 4500 m were mainly enriched in amino acid metabolism, lipid metabolism, and metabolism of xenobiotics by cytochrome P450, and they were also related to microbiota. Further analysis revealed that rumen metabolites accounted for 7.65% of serum metabolites. These common metabolites were mainly enriched in metabolic pathways and were significantly correlated with host genes (*p* < 0.05). This study found that microbiota, metabolites, and epithelial genes were coenriched in pathways related to lipid metabolism, energy metabolism, and immune metabolism, which may be involved in the regulation of Tibetan sheep adaptation to plateau environmental changes.

## 1. Introduction

Tibetan sheep is a unique dominant breed on the Qinghai–Tibet Plateau, and it evolved from wild sheep after long-term adaptation. The current livestock inventory exceeds 50 million [1]. They are mainly distributed throughout the Qinghai–Tibet Plateau and its adjacent areas at an altitude of 2500 m~4500 m and have good adaptability to the extreme environmental conditions of the Qinghai–Tibet Plateau, such as high altitude, low oxygen, cold, strong ultraviolet light, and nutritional stress. Compared with other sheep breeds, Tibetan sheep have a unique plateau adaptability mechanism. Tibetan sheep have been grazing on plateau pastures for generations, obtaining nutrients from natural herbage. This provides them with the energy needed for survival and production through a powerful rumen fermentation function, enabling them to adapt to the extreme and harsh plateau environment [2].

The rumen evolved approximately 40 million years ago [3] and is a hallmark organ of ruminants, playing an important role in host metabolism, immunity, and health [4]. The rumen contains a large number of complex microbiotas, including bacteria, archaea, fungi, and protozoa, which can digest proteins, starches, sugars, and plant cellulose and produce short-chain fatty acids (SCFAs), microbial proteins, ammonia, and other unknown metabolites [5]. These metabolites provide the energy and proteins required by the host body [6,7]. Moreover, they are involved in the regulation of key host functions, including nutrient processing, maintenance of energy balance, and immune system development [8,9]. Ultimately, these processes affect the production performance of ruminants. Ruminal function has a significant impact on the production system of ruminants. In high-altitude areas, the adaptive evolution of ruminants can influence genes related to energy metabolism, which, in turn, is related to rumen metabolism [10]. Food digestion and absorption are key processes in the adaptive evolution of animals. In addition to the role of the animal’s own genome, the intestinal symbiotic microbiome also plays an important role in helping the host digest food and synthesize nutrients that cannot be produced by the host itself. This, in turn, expands the metabolic reservoir of the animal host [11,12]. In fact, the rumen microbiota undergoes adaptive changes to the harsh plateau environment, and the intestinal microbiota and its function may be closely related to the adaptability of hosts at high altitudes on the Qinghai–Tibet Plateau [4,13]. Microbial genes in the rumens of high-altitude ruminants are coenriched in the metabolite VFA production pathway, providing persistent energy for the host. Additionally, the microbiome coevolves with the host genome to adapt to the extreme environment [10].Our previous studies also found differences in rumen microbiota abundance and composition of Tibetan sheep at different altitudes [14]. Host genes and microbiota are involved in the regulation of ruminal fermentation metabolism and epithelial immune barrier function. However, studies have shown that the phenotypic traits of ruminants are affected by rumen microbiota, and these effects can be observed through rumen metabolites [15]. The plateau adaptation of Tibetan sheep is a complex biological process closely related to all aspects of function and metabolism. Currently, studies on plateau adaptation only focus on physiological and biochemical blood indices, histomorphology, and gene expression, and only a few studies have focused on coevolutionary microbiota and their metabolites [10]. Metabolomics can reveal the metabolic phenotypes of animals [16] and microbiota [17]. Under heat stress, 13 potential metabolite markers related to carbohydrate, amino acid, and lipid metabolism in dairy cows are enriched, suggesting the impact of environmental stress on the metabolism of ruminants [18]. An increasing number of studies have demonstrated that comprehensively sequencing the intestinal microbiome and metagenome and correlating these data with the host genome, transcriptome, and metabolic profile has become a prominent research strategy for uncovering the relevant rules and mechanisms of host physiological characterization [19]. Multiomics studies have shown that the rumen microbiome and its metabolome and the host metabolome jointly promote individualized production performance in dairy cows [20]. Combined transcript and microbiome analysis showed that nutritional intervention improved rumen function and promoted compensatory growth in growth-retarding yaks [21]. The interactions among the rumen microbiome, the rumen epithelial transcriptome, and microbial metabolites indicate that a highly active early microbiome regulates rumen development in newborn calves at the cellular level [22].

Therefore, we hypothesized that rumen microbiota and metabolites could affect host metabolism (reflected by the serum metabolome) through the rumen epithelium, thus affecting Tibetan sheep adaptation to the plateau environment. In this study, the rumen microbiome and metabolome, host transcriptome, and serum metabolome of Tibetan sheep at different altitudes were studied, aiming to answer the following basic question: Do the rumen microbiota, metabolites, rumen epithelial-related gene expression, and host metabolites respond to plateau environmental changes? The differences in the rumen fluid metabolome and host metabolome in Tibetan sheep at different altitudes were compared to explore the regulation of rumen epithelium-related gene expression on metabolite transport and to reveal the influence of common metabolites on host adaptation to plateau environments. This study will reveal the mechanism of the effects of rumen microbes and metabolites on the host metabolic activities of Tibetan sheep at different altitudes, which will help improve the plateau adaptation performance of Tibetan sheep.

## 2. Results

### 2.1. Rumen Fluid Metabolic Profiles of Tibetan Sheep at Different Altitudes

A total of 1394 metabolites were identified in the rumen fluid of Tibetan sheep at the three altitudes. Principal component analysis (PCA) and cluster analysis revealed distinct differences in the metabolites among the three altitudes. FC > 2, *p* value < 0.01, and VIP > 1 were used as screening criteria for differential metabolite analysis. There were 194 differential metabolites between LA and MA, 216 differential metabolites between MA and HA, and 227 differential metabolites between LA and HA, among which 5 common differential metabolites were found between the three altitudes (Figure 1, Appendix A). Further screening of the up- and downregulated metabolites of the top 10 differential multiples (Appendix A), fructoselysine, 3–(2–Hydroxyphenyl) propionic acid, pantetheine 4′–phosphate, L–kynurenine, and phenethyl acetate, were found between LA and MA. However, 5’–ethylthioadenosine, venlafaxine, N–tigloylglycine, 5–methylthioribose, myristoleic acid, and fructoselysine were between MA and HA, etc. Between LA and HA, there were myricetin 3–O–glucoside, venlafaxine, tryptophyl–glutamate, L–kynurenine, and phenethyl acetate.

KEGG functional annotation found (Figure 2) that the differential metabolites of LA-MA and LA-HA were mainly annotated in amino acid metabolism, lipid metabolism, carbohydrate metabolism, and metabolism of cofactors and vitamins. Metabolites at medium and high altitudes above 3500 m were mainly annotated in amino acid metabolism and lipid metabolism. In LA-MA, upregulated differential metabolites were mainly enriched in vitamin B6 metabolism and the sphingolipid signaling pathway, and downregulated differential metabolites were mainly enriched in porphyrin metabolism and arachidonic acid metabolism. Between MA and HA, pyrimidine metabolism and purine metabolism were mainly enriched. In LA-HA, upregulated differential metabolites were mainly enriched in pyrimidine metabolism; downregulated metabolites were mainly enriched in glycerolipid metabolism.

### 2.2. Analysis of Rumen Microbiome–Metabolite Interactions in Tibetan Sheep at Different Altitudes

In a previous study, 19 phylum-level and 243 genus-level microbiotas were found between the three altitudes (Appendix A), and the interaction among the identified microbiota and metabolites was analyzed. According to the Procrustes analysis, differences were observed between rumen microbiota and metabolites at different altitudes (Figure 3). Furthermore, WGCNA dimension reduction analysis was conducted on metabolite data, the metabolites were divided into different metabolite modules, and correlation analysis was conducted with phylum-level flora (Figure 3). A total of 14 metabolite modules in the LA-MA formation were significantly correlated with Gemmatimonadetes, Acidobacteria, and Synergistetes (*p* < 0.05); 11 modules in MA-HA were significantly correlated with Fibrobacteres and Synergistetes (*p* < 0.05); and 12 modules in LA-HA were significantly correlated with Kiritimatiellaeota, Fibrobacteres, and Acidobacteria (*p* < 0.05). 

Correlation analysis of differential metabolites and differential microbiota (at the genus level) found that there was a strong correlation between the differential metabolites at different altitudes and the microbiota at the genus level (Appendix A). Screen|CC| > 0.8 and CCP < 0.05 data and the differential metabolites/differential microbiota of the top 30 frequencies were made into a correlation chord diagram (Figure 4). A total of 19 differential metabolites were found to be associated with 14 differential microbiotas between LA and MA. For example, protoporphyrin IX was strongly correlated with *Anaeroplasma*, *Shuttleworthia*, *Lysobacter*, *Pedomicrobium*, *Solirubrobacter*, etc. (*p* < 0.05); megastigmatrienone was strongly related to *Phyllobacterium*, *Lysobacter*, and *Lachnoclostridium_1* (*p* < 0.05); leukotriene D4 was strongly correlated with *Solirubrobacter* and *Pedomicrobium* (*p* < 0.05); and distearoyl phosphatidate was strongly correlated with *Lachnospiraceae_FE2018_group*, *Lysobacter*, and *Solirubrobacter* (*p* < 0.05). Between MA and HA, ricinoleic acid was correlated with *Ruminococcaceae_UCG-007*, *Ruminococcaceae_UCG-004*, and *Fretibacterium* (*p* < 0.05); ferulic acid was positively correlated with *Butyrivibrio_2*, *Mailhella*, and *Tyzzerella_3* (*p* < 0.05); ferulic acid was negatively correlated with *Ruminococcaceae_UCG-007*, *Phyllobacterium*, and *Ruminococcaceae_UCG-004* (*p* < 0.05); *Fretibacterium* and *Ruminococcaceae_UCG-007* were positively correlated with alpha-linolenic acid (*p* < 0.05); and *Lachnospiraceae_NK3A20_group* and *Mailhella* were negatively correlated with alpha-linolenic acid (*p* < 0.05). Between LA-HA, *Anaeroplasma* was positively correlated with butyryl-CoA (*p* < 0.05) and negatively correlated with alpha-linolenic acid (*p* < 0.05).

### 2.3. Analysis of Rumen Fluild Metabolome and Epithelial Transcriptome (mRNA) Interaction in Tibetan Sheep

A total of 27,916 expressed genes were detected in the rumen epithelia of Tibetan sheep at three altitudes by transcriptome sequencing, and 2184 differentially expressed genes were identified (Appendix A). Based on principal component analysis (PCA), significant differences were found among different altitudes, but there were no significant differences between MA and HA. WGCNA dimensionality reduction analysis of transcriptome and metabolome data revealed some correlation between gene modules and metabolite modules (Appendix A). By comparing the pathways involved in genes in the transcriptome and those involved in metabolites in the metabolome, we could obtain the number of coinvolved pathways (Figure 5). A common pathway (ko05310 asthma, Figure 6) was found between LA and MA, 7 common pathways were found between MA and HA, and 24 common pathways were found between LA and HA. The top 10 KEGG pathways identified in this study with the highest amount of coparticipation of differential genes and differential metabolites were counted for visual analysis. The differentially expressed genes and metabolites between LA and MA were coenriched in asthma, arachidonic acid metabolism, and biosynthesis of amino acids. Between MA and HA, differentially expressed genes and metabolites were mainly enriched in butanoate metabolism, propanoate metabolism, and biosynthesis of amino acids. Between LA and HA, differentially expressed genes and metabolites were mainly enriched in fatty acid degradation, propanoate metabolism, butanoate metabolism, pentose and glucuronate interconversions, and the cAMP signaling pathway.

### 2.4. Serum Metabolic Profiles of Tibetan Sheep at Different Altitudes

A total of 3582 metabolites were identified in the serum metabolic profiles of Tibetan sheep at the three altitudes. Principal component analysis (PCA) and cluster analysis showed that there were certain differences in serum metabolite composition at the three altitudes. By using the screening criteria of FC > 2, *p* value < 0.01, and VIP > 1, we identified 290 differential metabolites between LA and MA, 894 between MA and HA, and 543 between LA and HA (Appendix A), while 17 common differential metabolites were found between the three altitudes (Figure 7). Screening the up- and downregulated metabolites of the top 10 differential multiples (Appendix A), adenosine 2′,3′–cyclic phosphate, arginyl–lysine, glycyl–phenylalanine, hexyl 3–mercaptobutanoate, 1–(9Z–hexadecenoyl)–glycero–3–phosphate, and soyasaponin aa were found between LA and MA. Between MA and HA, there were vestitone 7–glucoside, isopropyl 3–(3,4–dihydroxyphenyl)–2–hydroxypropanoate, adenosine 2′,3′–cyclic phosphate, and gingerol. Between LA and HA, there were 8–epidiosbulbin E acetate, vestitone 7–glucoside, 3–O–sulfogalactosylceramide (d18:1/20:0), and yucalexin B′11.

KEGG functional annotation analysis of serum metabolites showed that LA-MA differential metabolites were mainly annotated in amino acid metabolism and lipid metabolism (Figure 8). Upregulated metabolite enrichment was found in the metabolism of xenobiotics by cytochrome P450 and nicotinate and nicotinamide metabolism, and downregulated metabolites were mainly enriched in galactose metabolism. In MA-HA, the main annotations were in amino acid metabolism and the digestive system, and the upregulated metabolites were concentrated in purine metabolism and dopaminergic synapses, while the downregulated metabolites were enriched in lysine biosynthesis and glycine, serine, and threonine metabolism. In LA-HA, the main annotations were in lipid metabolism; upregulated metabolites were enriched in sphingolipid metabolism, dopaminergic synapses, and vitamin B6 metabolism, and the downregulated metabolites were enriched in riboflavin metabolism and arachidonic acid metabolism.

### 2.5. Correlation Analysis of Rumen Microbes, Metabolites, and Serum Metabolites in Tibetan Sheep

WGCNA revealed that the serum metabolites of the three comparison groups were divided into 15, 10, and 11 metabolite modules, respectively, and these modules were correlated with microbiota to a certain extent. Further correlation analysis was carried out on the differential serum metabolites and differential microbiota (genus level). From the heatmap, it can be found that the differential serum metabolites at different altitudes were strongly correlated with the microbiota at the genus level (Appendix A). Using the screen|CC| > 0.8 and CCP < 0.05 data, the differential serum metabolites/differential microbes of the top 30 frequencies were made into a correlation chord diagram (Figure 9). Between LA and MA, 21 serum metabolites were found to be associated with 11 microbiotas. Specifically, 6-hydroxynicotinic acid and cysteinyl-proline were positively correlated with *Lysobacter*, *Pedomicrobium,* and *Solirubrobacter* (*p* < 0.05), while they were negatively correlated with *Lachnoclostridium_1* (*p* < 0.05); asparaginyl-glutamine was negatively correlated with *Lachnoclostridium_1* (*p* < 0.05) and positively correlated with *Phyllobacterium* and *Solirubrobacter* (*p* < 0.05). Between MA and HA, 20 metabolites were found to be correlated with 11 microbiota. Specifically, cytochalasin B was negatively correlated with *Mailhella* and positively correlated with *Ruminococcaceae_UCG-007* (*p* < 0.05). 13,14-dihydro-lipoxin A4 was positively correlated with *Butyrivibrio_2* and *Mailhella* (*p* < 0.05) and negatively correlated with *Fretibacterium*, *Ruminococcaceae_UCG-004,* and *Ruminococcaceae_UCG-007* (*p* < 0.05). Aspartic acid was negatively correlated with *Fretibacterium*, *Quinella*, *Ruminococcaceae_UCG-004,* and *Ruminococcaceae_UCG-007* (*p* < 0.05) and positively correlated with *Mailhella* (*p* < 0.05). Between LA and HA, 15 metabolites were found to have certain correlations with 11 microbiota, and biochanin A was found to be positively correlated with *Lysobacter*, *Ochrobactrum*, *Phyllobacterium,* and *Solirubrobacter* (*p* < 0.05).

Further comparative analysis of rumen fluid metabolites and serum metabolites showed that 274 rumen fluid metabolites were consistent with serum metabolites, accounting for 7.65% of the total host metabolites. There were 3 common differential metabolites between LA and MA, 10 common differential metabolites between MA and HA, and 6 common differential metabolites between LA and HA (Appendix A). In LA-MA, methyl cinnamate was upregulated in both rumen and serum metabolites, while geldanamycin was upregulated in serum and downregulated in rumens. In MA-HA, methyl cinnamate, lysyl-methionine, and embelin were downregulated in HA serum and rumens, while coniine and inosine were upregulated in high-altitude serum and rumens. In MA-HA, biochanin A was upregulated in HA serum and downregulated in rumens, while coniine was upregulated in both, and alpha-linolenic acid was downregulated in serum and upregulated in rumens. Further functional analysis of these common metabolites (Figure 10) showed that six different metabolites were enriched in 11 metabolic pathways. The metabolic pathway was mainly enriched, followed by the alpha-linolenic acid metabolism (ko00592) pathway. A correlation analysis was conducted between these metabolites and the genes involved in their respective pathways, revealing a certain level of correlation between the metabolites and the genes coenriched in those pathways. In the comparison between MA and HA, it was found that diethanolamine was positively correlated with the *GPCPD1* and *GPD1* genes, and alpha-linolenic acid was negatively correlated with the *PECR* gene and positively correlated with the *LOC101107420* gene. In the comparison of LA and HA, 3-(3-hydroxyphenyl) propanoic acid was found to be positively correlated with the *MOAO* gene.

## 3. Discussion

Previous studies on plateau animals have mainly focused on the genome level and physiological and biochemical levels of blood, lacking the perspective of microbes, metabolites, and host interactions. In this study, by integrating rumen microbes, rumen fluid metabolomes, serum metabolomes, and host transcriptomes, we studied the mechanism of influence of rumen microbes, metabolites on host genes, and the metabolic activities of Tibetan sheep at different altitudes to reveal the response of Tibetan sheep to plateau environmental changes. This study found certain differences in the rumen fluid metabolites of Tibetan sheep at different altitudes. Fructoselysine in MA increased significantly, while fructoselysine in LA and HA decreased. Fructoselysine can be converted to butyric acid [23], which is the main energy substance in ruminants and is involved in ketogenic metabolism [24]. These findings suggest that Tibetan sheep in the 3500 m area are more suitable for grazing and breeding, and similar results were also found in our previous studies [14]. The differential metabolite L–kynurenine decreases with altitude, and studies have found that kynurenine is associated with hypoxia and ischemia in animals [25]. Additionally, this study found that tryptophyl–glutamate significantly increased in HA, and glutamate plays a key role in nutrition, metabolism, and signal transduction [26]. These findings suggest that high-altitude Tibetan sheep may participate in the regulation of nutritional metabolism through amino acid metabolites. Further KEGG functional analysis showed that metabolites were mainly enriched in amino acid metabolism, lipid metabolism, carbohydrate metabolism, and metabolism of cofactors and vitamins (Figure 11). Most of the metabolites in the rumen were amino acids, peptides, and analogs, which was consistent with previous reports [27,28]. These amino acids are key precursors for the synthesis of proteins and peptides [29]. In addition, lipid metabolism in the rumen is also very active [30], primarily involved in regulating the antibacterial effect of fatty acids and microbial hydrogenation, as well as changing the absorption of fatty acids to improve production performance [31]. These findings indicate that Tibetan sheep may pass lipid metabolism to improve their growth performance. However, in the comparison between low altitude and middle altitude, upregulated metabolites were mainly enriched in vitamin B6 metabolism. Vitamin B has been found to play an important role in immune regulation [32,33], suggesting that in a plateau environment of 3500 m, Tibetan sheep may enhance their immunity through vitamin B6 metabolism. From the vitamin B6 metabolism pathway diagram (ko00750), upregulated metabolites are also involved in glyoxylate and dicarboxylate metabolism, butanoate metabolism, and the pentose phosphate pathway, which regulate pathways related to energy metabolism in response to extreme environments (Appendix A). At the high altitude of 4500 m, the upregulated metabolites are enriched in pyrimidine metabolism and participate in the energy process of cellular ATP [34,35] (Appendix A), which plays a crucial role in regulating the phenotypic traits of Tibetan sheep to facilitate adaptation to the plateau environment. It was found that the phenotypic traits of ruminants were affected by rumen microbiota, and their functions could be reflected by rumen metabolites [15]. Our previous studies have confirmed that there are differences in microbiota at different altitudes [14]. Further combined analysis of the microbiome and metabolome revealed that Fibrobacteres and Synergistetes were significantly correlated with metabolites. Fibrobacteres are considered to be the main bacterial degraders of lignocellulosic substances in the rumens of herbivores [36], which enables them to degrade herbage in a high-altitude environment and produce energy metabolites. Protoporphyrin IX and leukotriene D4 were found to be strongly correlated with Solirubrobacter in the comparison between low altitude and middle altitude. Protoporphyrin IX is a precursor of heme, which is related to hypoxic regulation [37,38]; leukotriene D4 is related to inflammation in the body [39]; and leukotriene D4 content is decreased at high altitude compared with low altitude. Ricinoleic acid and ferulic acid are correlated with *Ruminococcaceae* and *Butyrivibrio_2* at medium and high altitudes, and *Ruminococcaceae* can degrade cellulose and starch [40,41]. The number of *Butyrivibrio* is positively correlated with the number of methane bacteria [42]. Meanwhile, ricinoleic acid is correlated with bactericide and anti-inflammatory effects [43], and ferulic acid is correlated with cardiovascular diseases [44]. Metabolites linked to these diseases show correlations with these bacteria, which may contribute to improving the immunity of Tibetan sheep. *Anaeroplasma* is positively correlated with butyryl-CoA at low and high altitudes. *Anaeroplasma* is positively correlated with body obesity [45]. In the plateau environment, butyryl-CoA is regulated by Anaeroplasma, which affects butyric acid synthesis [46]. These results indicate that the content of metabolites is influenced by the abundance of microbiota, and thus it affects the phenotypic traits of the organism. 

How are rumen metabolites transported to the corresponding organs and tissues for regulation? We further analyzed the interaction between these metabolites and the host and found that there was a certain correlation between the gene module and the metabolite module of the rumen epithelium. The differentially expressed genes and metabolites between low altitude and high altitude were enriched in asthma, arachidonic acid metabolism, biosynthesis of amino acids, and other pathways. The asthma pathway regulates respiratory pathways (Figure 6), in which the *MHCII* gene is upregulated, thereby affecting immune indices and reducing the content of downstream metabolite leukotriene D4, which is a lipid mediator closely related to inflammation [47]. It also regulates metabolic immunity through the arachidonic acid metabolism pathway [48]. These results still need to be validated at the cellular level. However, compared with the high altitude of 4500 m, differential genes and differential metabolites were enriched in butanoate metabolism, propanoate metabolism, and fatty acid degradation. By reducing the expression of related genes (*ACADS*, *HMGCL*, *HMGCS2*, *EHHADH*, *ECHS1*, etc.) (Figure 11 and Appendix A), the content of butyryl-CoA downstream can be reduced, thus affecting the content of butyric acid in the rumens of high-altitude Tibetan sheep. This was consistent with our previous results [14], which also confirmed that rumen microbial metabolites interact with host genes to regulate the body pathway (Figure 11). In addition, differentially expressed genes and metabolites were enriched in the cAMP signaling pathway (Appendix A), which participates in the regulation of the PKA pathway by reducing the level of the metabolite dopamine, resulting in increased expression of the downstream gene *CREB5*, thus stimulating neurons, regulating body traits, and adjusting to the environmental pressure of plateau mutation. The specific regulation of plateau animal traits still needs subsequent verification at the cellular level. However, some metabolites in the rumen are transported into the blood through the epithelium to be subsequently transported to the corresponding tissues and organs to play a role. We further analyzed the serum metabolic profile to reveal the differences in phenotypic traits. There were also differences in serum metabolite composition among the three altitudes, and adenosine 2’,3’–cyclic phosphate was found to be enriched in the purine metabolism (ko00230) pathway at low altitudes compared with medium and high altitudes. Purines are important factors in cell proliferation and immune regulation [49]. The adenosine 2’,3’–cyclic phosphate contents were increased at a medium altitude of 3500 m to regulate the immunity of Tibetan sheep. KEGG functional annotation analysis of differential metabolites showed that differential metabolites at middle and high altitudes were mainly enriched in amino acid metabolism and lipid metabolism, which is consistent with the previous functional enrichment of microbial metabolites. It is mainly related to the synthesis of amino acids of some proteins and lipid metabolism, thus improving production performance [31]. Some upregulated metabolites are mainly enriched in the metabolism of xenobiotics by cytochrome P450 and nicotinate and nicotinamide at the transition from low altitude to 3500 m. These metabolites participated in the regulation of immune metabolism-related pathways [50,51], improving the immunity of Tibetan sheep. Whereas the downregulated metabolite enrichment is in galactose metabolism, galactose related to energy metabolism decreases with altitude [52]. However, from the middle altitude of 3500 m to the high altitude of 4500 m, different metabolites are mainly enriched in amino acid metabolism and the digestive system, and Tibetan sheep may need higher amino acid metabolism, digestion, and absorption capacity in the high-altitude environment. 

What is the relationship between serum metabolites and rumen microbiota? The conjoint analysis revealed that there was a significant positive correlation between the differential metabolite 6-hydroxynicotinic acid and some microbiota from plant soil [53,54] (*Lysobacter*, *Pedomicrobium*, and *Solirubrobacter*) between low and middle altitudes. 6-hydroxynicotinic acid is involved in regulating respiratory-related diseases and plays a positive role [55], with the highest content at the mid-altitude of 3500 m, which may be greatly related to this soil microbe at high altitude. In addition, it was found that cysteinyl-proline related to angiotensin converting enzyme was also significantly positively correlated with related microbiota extracted from soil, and its content was also the highest in the middle altitude. Cytochalasin B has been found to be significantly related to microbiota at medium and high altitudes. Cytochalasin B, a mycotoxin with cellular permeability, can inhibit the formation of actin microfilaments and participate in the regulation of fat [56]. Aspartic acid is significantly related to some bacteria and is an endogenous amino acid that plays an important role in the development of the neuroendocrine system and nervous system [57]. Biochanin A, which has anti-inflammatory, anticancer, neuroprotective, antioxidant, and antimicrobial properties [58], was also significantly correlated with microbiota when compared with those at low altitude and high altitude. The correlation analysis between serum metabolites and rumen microflora showed that there was a strong correlation between them, and further comparative analysis showed that 7.65% of the serum metabolites were rumen metabolites. At low altitude compared to medium altitude, methyl cinnamate is upregulated in serum and rumens, and it was found that methyl cinnamate can inhibit adipocyte differentiation by activating the CaMKK2-AMPK pathway in 3T3-L1 preadipocytes [59], indicating that adipocyte differentiation is regulated by the upregulation of metabolite contents in the 3500 m plateau environment, thus regulating body energy. At 4500 m, the methyl cinnamate content decreased, so we speculate that Tibetan sheep are more suitable for grazing and breeding at 3500 m. In addition, it was found that the content of inosine was upregulated in the serum and rumen at high altitudes, and inosine is an important secondary metabolite of purine metabolism and a molecular messenger in the cell signaling pathway [60]. The content of coniine, a metabolite harmful to the body, was found to increase at 4500 m at high altitudes [61], and it was also confirmed that 3500 m is the best elevation for Tibetan sheep grazing and breeding. Further functional analysis of common metabolites showed that they were mainly concentrated in metabolic pathways, followed by alpha-linolenic acid metabolism (ko00592). In addition, it was found that diethanolamine was related to *gene-GPCPD1* and *gene-GPD1* in the comparison between middle altitude and high altitude, and *gene-GPCPD1* was related to choline metabolism [62]. It was also found that alpha-linolenic acid, a metabolite associated with anticancer, anti-inflammatory, antioxidant, and antiobesity properties, as well as neuroprotection and regulation of intestinal flora [63], was negatively correlated with the *PECR* gene. In the rumen, the increased content of alpha-linolenic acid at high altitude may be involved in the regulation of intestinal flora, as well as anti-inflammatory effects, and studies have also found that the *PECR* gene regulates lipid metabolism [64], indicating an increase in alpha-linolenic acid at high altitude, and the lipid metabolism function of the rumen epithelium may be decreased. In addition, at low altitudes compared to high altitudes, it was found that 3-(3-hydroxyphenyl) propanoic acid was positively correlated with the *MOAO* gene, and the metabolite was enriched in phenylalanine metabolism (ko00360), which was found to reduce the occurrence of inflammation [65]. In conclusion, in response to changes in the high-altitude environment, Tibetan sheep regulate the immune level through the interaction of metabolites and genes. 

## 4. Methods

### 4.1. Test Design, Location, and Sample Collection

The animals involved in the research were subject to the Regulations of the Ministry of Science and Technology, PRC on the Administration of Laboratory Animal Affairs (PRC; revised in June 2004). The sample collection program was approved by the Ethics Committee of Animal Husbandry Specialty of Gansu Agricultural University (approval no. GAU-LC-2020-27). Plateau-type Tibetan sheep were selected as the study object (3.5 years old, ♀, *n* = 6/group), and all the experimental animals were in good health, in the nonpregnant stage, and in the same physiological state. During the same period, they ate natural grass freely without any supplementary feeding in the plateau environment. Forage species and chemical composition were described in the previous article [14]. Samples were collected from Zhuoni (LA, 2500 m), Haiyan (MA, 3500 m), and Yushu (HA, 4500 m) on the Qinghai–Tibet Plateau in August 2020 (Figure 12). Jugular blood was collected in vacuum tubes and centrifuged (5000× *g*, 20 min, 4 °C), and serum was isolated and stored at −20°C for serum metabolome determination. Rumen fluids were collected by a rumen vacuum sampler (pipe from the mouth into the rumen), and rumen fluids were divided into frozen tubes and stored in liquid nitrogen for subsequent 16S rRNA sequencing and metabolome determination. Subsequently, according to the traditional slaughter method, a sterilized knife was used to cut the jugular vein of the neck of the test sheep, and death was caused by instant bloodletting. Then, the abdominal cavity was dissected, the rumen organs were removed, and epithelial tissue samples from the ventral portion of the rumen were collected, divided into freeze-storage tubes, and stored in liquid nitrogen for subsequent RNA sequencing.

### 4.2. DNA Extraction and 16S rRNA Sequencing

An MN NucleoSpin 96 Soil kit (Macherey-Nagel, 740787.2, Düren, Germany) was used to extract microbial DNA from the rumen content samples, and the concentration and purity were measured by a NanoPhotometer-N60 (Implen, Munich, Germany). Through PCR amplification of the V3–V4 region of the 16S rRNA gene, the community structure of the rumen microbiota was obtained. The primers were 338F 5′-ACTCCTACGGGAGGCAGCAG-3′ and 806R 5′-GGACTACHVGGGTWTCTAAT-3′. The library obtained by this PCR amplification process was then sequenced on an Illumina MiSeq platform (Illumina, San Diego, CA, USA), and bioinformatics analysis was performed using BMKCloud (www.biocloud.net) (accessed on 5 September 2022). The raw data returned by the Illumina MiSeq platform were subjected to merging of paired-end reads. Using FLASH v 1.2.7 software, the reads of each sample were spliced according to the minimum overlap length of 10 bp and the maximum mismatch ratio allowed in the overlap region of 0.2, and the spliced sequence obtained was the original tag data (raw tags). Trimmomatic v 0.33 software was used to filter the raw tags obtained by splicing, and the parameter was set to a 50 bp window. If the average quality value in the window was lower than 20, the base at the back end was cut off from the window, and the tags whose lengths were less than 75% of the lengths of tags after quality control were filtered to obtain high-quality tag data. Finally, UCHIME v 4.2 software was used to identify and remove the chimeric sequences to obtain the final effective tags. Usearch v 11.0.1 software [66] was used to cluster tags at the 97% similarity level to obtain operational taxonomic units (OTUs), and the OTUs were annotated based on the Silva (bacteria) taxonomic database (https://www.arb-silva.de/) (accessed on 6 Sptember 2022). Based on the OTU analysis results, taxonomic analysis was performed on the samples at various taxonomic levels, and the community structure of the samples at the taxonomic levels of phylum, class, order, family, genus, and species was obtained. Then, a *t*-test was performed on the species abundance data between groups using Metastats V1 (http://metastats.cbcb.umd.edu/) (accessed on 6 Sptember 2022) software [67], the *p* value was obtained, and the q value was obtained by correcting the *p* value. Finally, the species leading to the composition difference between the two groups of samples were screened according to the q value.

### 4.3. Metabolic Spectrometry of LS-MS/MS

The metabolites in the rumen contents and serum of 18 Tibetan sheep were determined by liquid chromatography–mass spectrometry. The samples were defrosted at room temperature, pretreated according to the methods of Dunn et al. (2011) and Want et al. (2010) [68,69], and prepared by referring to the method of Liu et al. [70]. Finally, 10 μL was mixed into QC samples for machine detection. The detection platform was a Waters Acquity I-Class PLUS ultrahigh-performance liquid chromatography tandem Waters Xevo G2-XS QTOF high-resolution mass spectrometer. Chromatography was performed on an Acquity UPLC HSS T3 column (1.8 µm, 2.1 × 100 mm) purchased from Waters. The sample detection parameters were determined by Liu et al. [70]. The original data collected by MassLynx V4.2 were used for peak extraction, peak alignment, and other data processing operations by Progenesis QI v2.0 software. The identification was carried out based on the online METLIN database, a public database, and the BMK self-built database of Progenesis QI v2.0 software, and theoretical fragment identification was carried out. Parent ion mass number deviation of 100 ppm, fragment ion mass number deviation of 50 ppm, and bioinformatics analysis of the identified metabolites was performed on BMKCloud (www.biocloud.net) (accessed on 3 October 2022), and the differential metabolites were screened by combining the difference multiple, the *p* value of the *t*-test, and the VIP value of the OPLS-DA model. The screening standard was fold change (FC) > 1, *p* value < 0.05, and variable importance in projection (VIP) > 1. Kyoto Encyclopedia of Genes and Genome (KEGG) functional annotation and enrichment analyses were performed for differential metabolites [71].

### 4.4. RNA Extraction and Transcriptome Sequencing

TRIzol reagent (Invitrogen, DP662-T1C, CA, USA) was used to extract total RNA from rumen epithelial tissue of Tibetan sheep. The Nanodrop2000 (v1, Thermo, Beijing, China) was used for concentration detection. Agient2100, LabChip GX (platinum, the model Platinum Elmer LabChip GX), was tested for integrity. The cDNA Library was constructed by a VAHTS Universal V6 RNA-seq Library Prep Kit for an Illumina^®^ kit (Vazyme, Nanjing, China, NR604-02) in strict accordance with the kit procedure. A VAHTSTM DNA Clean Beads Kit (Vazyme, Nanjing, China, N411-03) was further used for product purification. The constructed library was sequenced using an Illumina NovaSeq6000 (San Diego, CA, USA). Bioinformatics analysis was performed on BMKCloud (www.biocloud.net) (accessed on 20 October 2022), and clean data were obtained after data filtering. Sequence alignment between clean data and the specified reference genome, Ovis_aries (Oar_rambouillet_v1.0.Ovis_aries), was performed by HISAT (v 0.1.6) [72]. Mapped data were obtained (using an indexing scheme based on the Burrows–Wheeler transform and the Ferragina–Manzini (FM) index, using two types of indices for alignment), and StringTie (The Center for Computational Biology at Johns Hopkins University, Baltimore, Maryland, USA) [73] was used to assemble reads on pairs. FPKM [74] (fragments per kilobase of transcript per million fragments mapped) was used as an indicator to measure the level of transcript or gene expression. The DESeq2 data analysis method was used for differential expression gene analysis (FC), representing the ratio of expression between two samples (groups), and the false discovery rate (FDR) was obtained by correcting the differential significance *p* value. FC > = 2 and FDR < 0.01 were used as the screening criteria for differentially expressed genes.

### 4.5. Data Analysis

According to McHardy et al.’s method [75], conjoint analysis of the microbiome and metabolome was conducted, and the R (R-3.6.1) tool was used for subsequent informatics analysis. PCoA (principal coordinate analysis) was selected to conduct dimensionality reduction sequencing of the microbiome (genus level) and metabolome (R vegan package). First, the distance matrix was calculated by the quantitative matrix of microbiota and metabolites. The distance algorithm of the microbiome was Bayesian distance, the distance algorithm of the metabolome was Euclidean distance, and the distance sorting was carried out by PCoA. The coordinates of characteristic axes in the PCoA results of the microbiome and metabolome were extracted, and Procrustes analysis was carried out to compare the similarity and variation between the microbiome and metabolome (using the vegan package). Metabolic data were dimensionally reduced by weighted gene coexpression network analysis (WGCNA), and metabolites were divided into different metabolite clusters (using the WGCNA package). The expression of metabolite clusters was represented by the median content of the same cluster. Pearson correlation analysis was carried out with microbiota, a heatmap was drawn, and correlation analysis results were screened (Hmisc package, pheatmap package). The screening condition was evaluated, and its standard was: correlation coefficient *p* value (CCP) < 0.05, and then the occurrence frequency of metabolite clusters/microbiota was counted, and the correlation result table of metabolite clusters/microbiota with top 30 frequency was used to draw a chord diagram (circlize package). Using the R tool, the WGCNA package was used to perform WGCNA dimensionality reduction analysis on all metabolites and genes in the differential groups (when the number of genes or metabolites > 10000, low expression filtering was applied) [76], and genes and metabolites were divided into different modules. Through the Hmisc package for correlation analysis, the eigengene of the corresponding modules represented the gene/metabolome module content, and the correlation between the transcriptome module and metabolome module after dimension reduction was calculated. We retained *p* values with at least one set of correlations to satisfy CCP < 0.05 data through the pheatmap package for correlation map visualization and heatmap drawing. In addition, the pathways involved in genes in the transcriptome and those involved in metabolites in the metabolome were compared to obtain the number of coinvolved pathways. The Venn diagram was drawn using the Venn.diagram function in the Venn Diagram function package through the R language tools [77]. Through the ggplot2 package in R language tools, the top 10 KEGG pathways with the largest number of coparticipating genes and metabolites identified in this experiment were counted for visual analysis.

## 5. Conclusions

The purpose of this study was to elucidate the response of rumen microbiota and metabolites to host (mRNA and metabolite) interactions in the Tibetan sheep plateau environment. Compared with low altitudes, metabolites at medium and high altitudes were enriched in amino acid metabolism and lipid metabolism, which participate in the regulation of amino acid metabolism and fatty acid absorption, thereby affecting production performance. Especially at an altitude of 3500 m, upregulated metabolites were enriched in vitamin B6 metabolism and participated in the regulation of the body’s immune and energy metabolism processes, suggesting that Tibetan sheep were more suitable for grazing and breeding in an environment of approximately 3500 m. In addition, this study also confirmed the interaction between rumen metabolites, metabolites, and genes; the differentially expressed genes and metabolites in high-altitude areas were coenriched in asthma (ko05310), arachidonic acid metabolism, and biosynthesis of amino acids pathways and participated in the regulation of respiratory and immune-related functions to adapt to the plateau environment. Host metabolome analysis revealed that the same functional pathways as the rumen metabolome were enriched in amino acid metabolism, lipid metabolism, and metabolism of xenobiotics by cytochrome P450. The host metabolome is involved in the regulation of amino acid synthesis, lipid metabolism, and immune metabolism pathways. A total of 7.65% of common metabolites were found in the host serum and rumen, and methyl cinnamate was mainly enriched in the metabolic pathway. Methyl cinnamate increases at the middle altitude of 3500 m to participate in the regulation of adipocyte differentiation and thus energy metabolism of the body. Moreover, these metabolites are associated with some genes related to inflammation and lipid metabolism, suggesting that metabolites interact with host genes to regulate the immune level of the body, thus responding to plateau environmental changes.

## Figures and Tables

**Figure 1 ijms-24-14856-f001:**
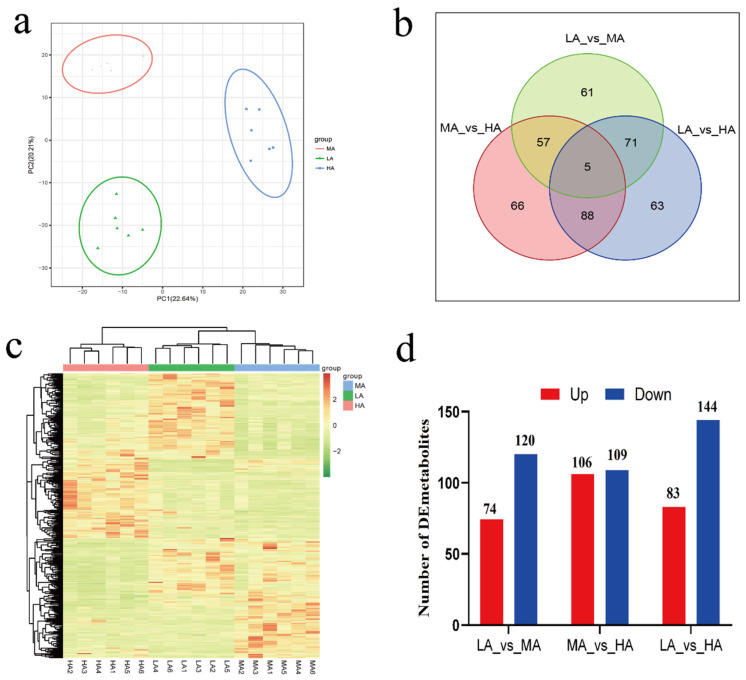
Ruminal microbial metabolism profile of Tibetan sheep at different altitudes. (**a**) PCA. (**b**) Veen map. (**c**) Clustering heat map. (**d**). Up and down metabolite statistics. The numbers on the Veen chart and bar chart represent the amount of metabolites.

**Figure 2 ijms-24-14856-f002:**
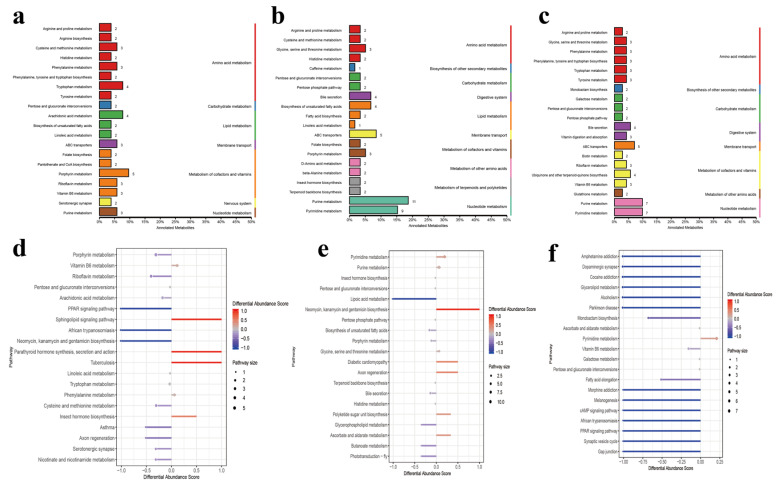
KEGG function analysis of microbial differential metabolites. (**a**–**c**) KEGG annotation of the classification diagram. (**d**–**f**) KEGG functional difference abundance score map. Note: (**a**,**d**) LA-MA. (**b**,**e**) MA-HA. (**c**,**f**) LA-HA. DA score reflects the overall changes of all metabolites in the metabolic pathway, A score of 1 indicates that the expression trend of all annotated metabolites in this pathway is upregulated, while −1 indicates that the expression trend of all annotated metabolites in this pathway is downregulated. The number on the column represents the amount of differential metabolite annotated by the pathway. The larger the dot, the more metabolites there are.

**Figure 3 ijms-24-14856-f003:**
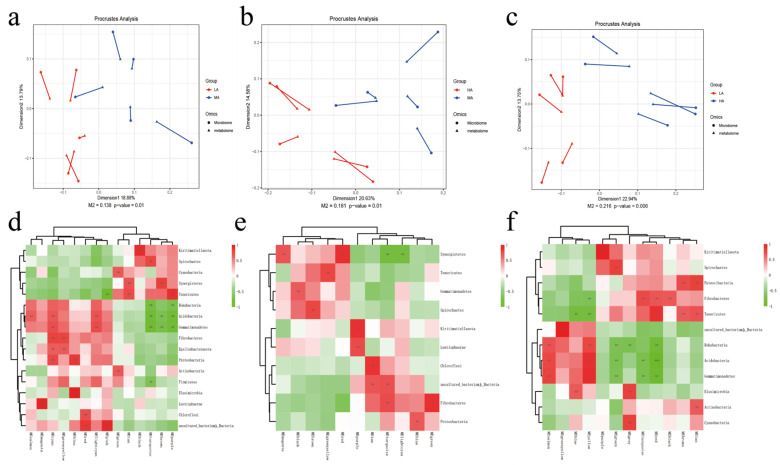
Microbial–metabolite correlation analysis. (**a**–**c**) Microbial–metabolite Procrustes analysis. (**d**–**f**) Microbial–metabolite correlation heat map. Note: (**a**,**d**) LA-MA. (**b**,**e**) MA-HA. (**c**,**f**) LA-HA.** and *** indicate, respectively, *p* < 0.01, and *p* < 0.001.

**Figure 4 ijms-24-14856-f004:**
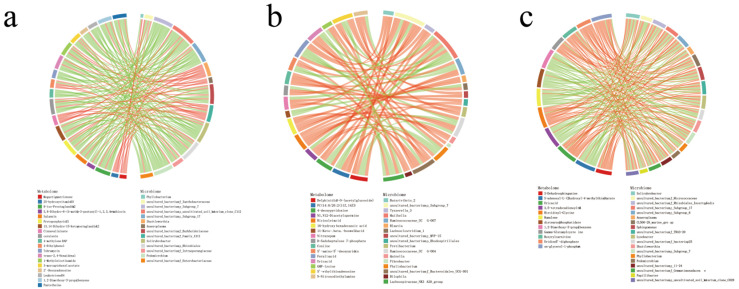
Correlation chords of metabolites/different microbiota at different altitudes. Note: The red strings represent a positive correlation, and the green strings represent a negative correlation. (**a**) LA-MA. (**b**) MA-HA. (**c**) LA-HA.

**Figure 5 ijms-24-14856-f005:**
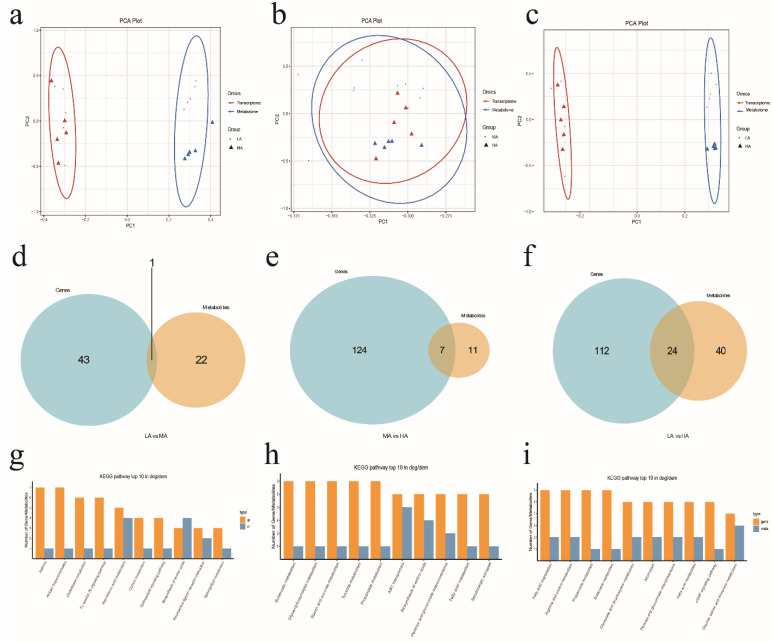
Microbiota metabolome interacting with epithelial transcriptome (mRNA). (**a**–**c**) PCA analysis. (**d**–**f**) Venn diagram of differential genes and differential metabolite pathways. (**g**–**i**) The top 10 pathways that contain the most differential genes/differential metabolites. (**a**,**d**,**g**) LA-MA. (**b**,**e**,**h**) MA-HA. (**c**,**f**,**i**) LA-HA. The numbers on the Venn map represent the number of functional pathways where differential genes and differential metabolites are enriched.

**Figure 6 ijms-24-14856-f006:**
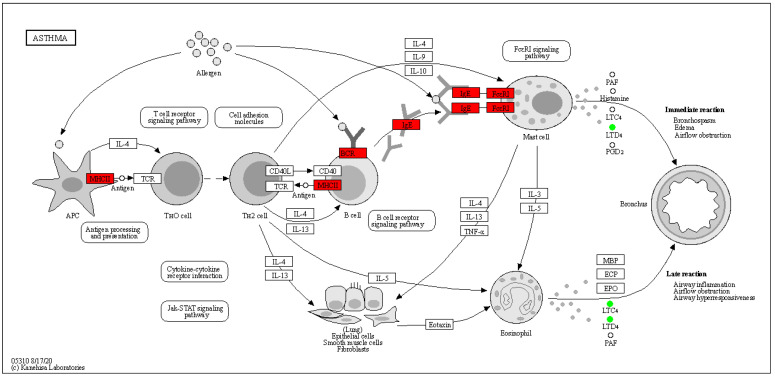
Asthma pathway. Note: Red represents upregulated genes and green represents downregulated metabolites.

**Figure 7 ijms-24-14856-f007:**
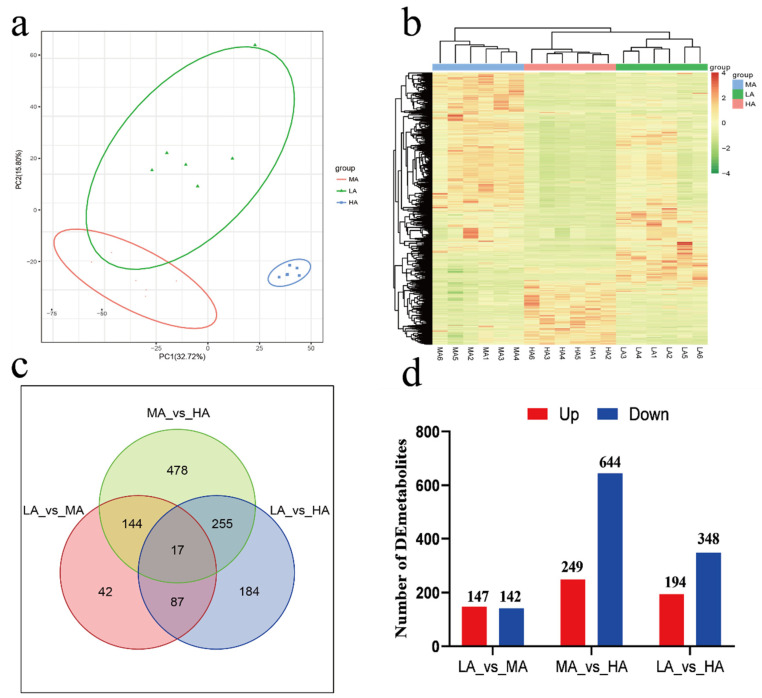
Serum metabolism profile of Tibetan sheep at different altitudes. Note: (**a**) PCA. (**b**) Venn map. (**c**) Clustering heat map. (**d**) Up and down metabolite statistics. The numbers on the Veen chart and bar chart represent the amount of metabolites.

**Figure 8 ijms-24-14856-f008:**
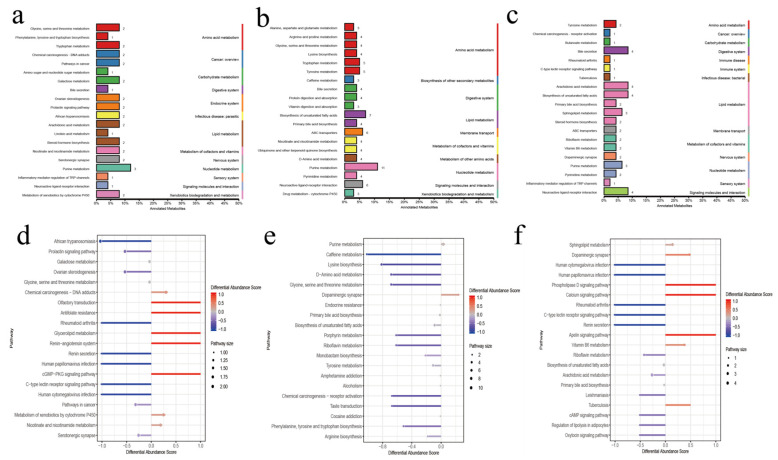
KEGG function analysis of serum metabolites. (**a**–**c**) KEGG annotation of the classification diagram. (**d**–**f**) KEGG functional difference abundance score map. Note: (**a**,**d**) LA-MA. (**b**,**e**) MA-HA. (**c**,**f**) LA-HA. The number on the column represents the amount of differential metabolite annotated by the pathway.

**Figure 9 ijms-24-14856-f009:**
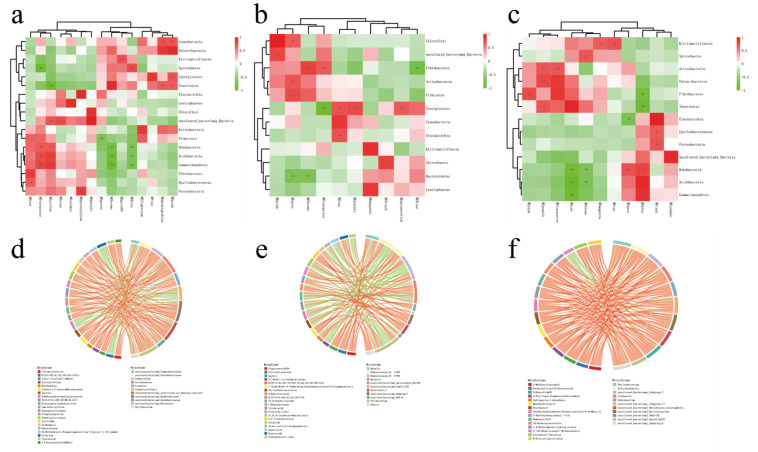
Correlation analysis between serum metabolites and microbiota. (**a**–**c**) Microbial-metabolite correlation heat map. (**d**–**f**) Correlation chords of metabolites/different microbiota at different altitudes. Note: (**a**,**d**) LA-MA. (**b**,**e**) MA-HA. (**c**,**f**) LA-HA. ** and *** indicate, respectively, *p* < 0.01, and *p* < 0.001. The red strings represent a positive correlation, and the green strings represent a negative correlation.

**Figure 10 ijms-24-14856-f010:**
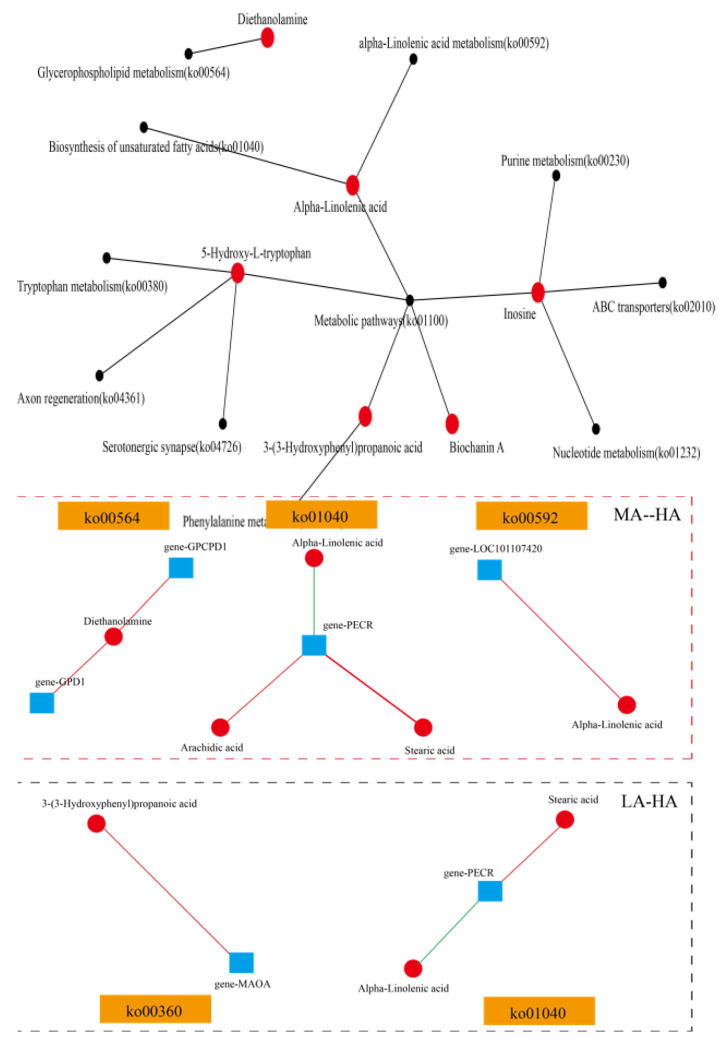
Functional analysis of common metabolites and correlation analysis with genes. Note: The red lines represent positive correlations, and the green lines represent negative correlations. The red dots represent metabolites and the black dots represent metabolic pathways.

**Figure 11 ijms-24-14856-f011:**
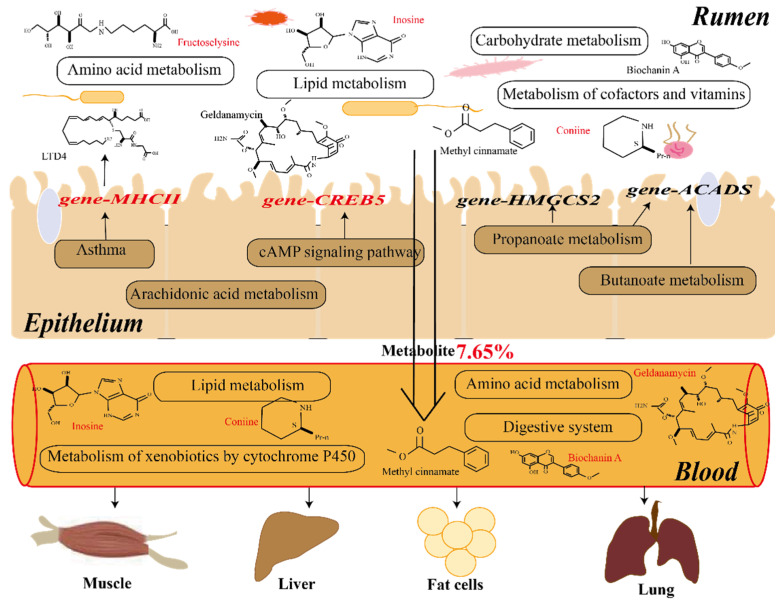
Microbes and the metabolite–mRNA–serum metabolite interaction model. Note: The red font represents upregulation of metabolites and genes, and the black font represents downregulation. The epithelium functions as a pathway for coenrichment of microbial metabolites and host mRNA. Red metabolites and gene representatives are upregulated at high altitudes, and black represents downregulation.

**Figure 12 ijms-24-14856-f012:**
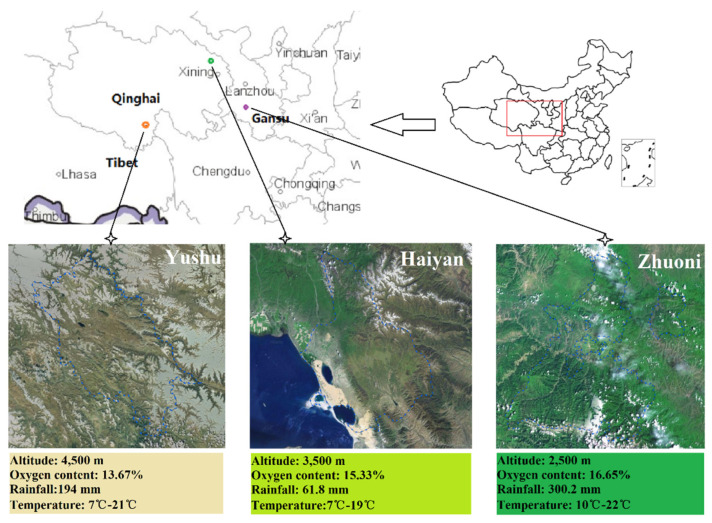
Qinghai–Tibet Plateau sampling sites. (Source: Sha, et al. [14].) Environmental information (altitude, oxygen content, rainfall, temperature) at three different altitude of the Tibetan Plateau.

## Data Availability

The datasets presented in this study can be found in online repositories. The names of the repositories and accession numbers can be found below: Sequence Read Archive (SRA): PRJNA818841 and PRJNA819418.

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
