# Peer review of "Synergistic Responses of Tibetan Sheep Rumen Microbiota, Metabolites, and the Host to the Plateau Environment"

_ijms, 2023, doi:10.3390/ijms241914856_

Round 1
Reviewer 1 Report
The title is illogical. The word "metabolites" was mentioned twice in the single sentence, which is confusing.
The authors ran a host tissue transcriptome analysis. The expression data of genes at mRNA levels are not equivalent of host genes; and the latter usually means the structure and function at genomic DNA levels.
The concept errors:
1. The rumen has several sacs: dorsal or ventral blindsac, ventral, and cranial sacs. The reviewer does not understand what the abdominal sac of the rumen means.
2. The metabolites from the rumen contents are not solely microbial origin and can come from the diet and host metabolism. As a result, the authors should not use "microbial metabolites" to represent the metabolites of the rumen content without further analysis.
The lack of details:
The authors listed several data tools used but failed to provide any details (software names and version, parameter setting, and input data and metadata), such as WGCNA, Conjoint analysis
The data provided do not support the main conclusion that "microbiota, its metabolites and host metabolites jointly act on rumen epithelial gene expression pathways”.
The data provided do not indicate any causal relationship. It is possible that host gene expression affects host metabolites.
How was "bloodletting of jugular vein" done? jugular venipuncture?
The info on vendors or suppliers of the reagents used was not provided, such as VAHTS Universal V6 RNA-seq Library Prep Kit for Illumina® kit (NR604-02).
The manuscript is full of grammatical errors.
The manuscript is poorly prepared with numerous grammatical errors.
Author Response
Please see the attachment.
I am very grateful to the reviewers for spending their valuable time to review my research paper and making useful suggestions, which are of great help to improve the quality of the paper. We revised the full paper according to the comments of the reviewers, see the revised manuscript for details.

Reviewer 2 Report
The abstract should be a total of about 200 words maximum.
Be careful along the text: often instead of a period there is a semicolon followed by a new sentence with a capital letter
Methods: for all reagents add catalogue number and specify the company
Paragraph 4.1
Explain the statistical test that led to the sampling of 6 items, explain why only females were used
and whether they had previously been pregnant
insert a geographical map indicating the three different places, the rainfall and above all the chemical composition of the fresh food
better explain the abdominal sac
Moderate editing of English language required
Author Response
I am very grateful to the reviewers for spending their valuable time to review my research paper and making useful suggestions, which are of great help to improve the quality of the paper. We revised the full paper according to the comments of the reviewers, see the revised manuscript for details. The following are responses to comments made by the reviewers.
Comments and Suggestions for Authors
The abstract should be a total of about 200 words maximum.
Reply: Thank you for your comments. We have simplified and revised the abstract part.
Be careful along the text: often instead of a period there is a semicolon followed by a new sentence with a capital letter
Reply: Thank you for your suggestions. The whole article has been revised.
Methods: for all reagents add catalogue number and specify the company
Reply: Thank you reviewer for your questions. We have revised the material methods section and added relevant details.
Paragraph 4.1
Explain the statistical test that led to the sampling of 6 items, explain why only females were used
and whether they had previously been pregnant
Reply: Thank you reviewer for your questions. In this study, we selected 6 ewes at each altitude. The random sampling method was adopted, and 6 ewes aged 3.5 years were randomly selected from the same breeding group to ensure that the samples within the group had higher repeatability and smaller differences, which could represent the real situation of the group. In addition, it is also seen in our result analysis that the samples at each sampling point are clustered very well.
The reason for choosing female sheep in this study is that Tibetan sheep in plateau is an important germplasm resource of sheep in plateau area, so the study of its backup female sheep group is helpful to the subsequent propagation of Tibetan sheep group. The 3.5-year-old ewes selected in this study were in the non-pregnant period during the sampling period, which ensured that their physiological conditions were consistent. We have also made modifications and supplements in the revised draft.
insert a geographical map indicating the three different places, the rainfall and above all the chemical composition of the fresh food
Reply:Thank you for your suggestions. We have added geographical maps in the modification, indicating altitude, oxygen content, rainfall, temperature, etc. For the species and chemical composition of the forage, we have published relevant information in a previous article, and the forage information is cited in this article.
better explain the abdominal sac
Reply: I'm sorry, we're using the wrong terminology. The rumen has several sacs: dorsal or ventral blindsac, ventral, and cranial sacs. The points we collected were ventral parts belonging to the rumen. The modifications were made in the article.
Comments on the Quality of English Language
Moderate editing of English language required
Reply: The language grammar and other issues of the article have been reviewed and revised by native English experts.

Round 2
Reviewer 2 Report
many thanks for replying to my request
Author Response
We thank the reviewers for their review and replies to the revised manuscript.